# Effect of cystic fibrosis transmembrane conductance regulator modulators and dedicated cystic fibrosis gastrointestinal clinic visits on the incidence of distal intestinal obstructive syndrome in persons with cystic fibrosis

**Nicha Wongjarupong**[1,2*‡], **Malique F. Delbrune**[3‡], **Jameel Alp**[4‡], **Daphne M. Moutsoglou**[4,5], **Talia Wiggen**[6], **Ashley Benner**[6], **Joanne L. Billings**[7], **Jordan M. Dunitz**[7], **Sarah J. Schwarzenberg**[8], **Baharak Moshiree**[9], **Shahnaz Sultan**[1]

**1** Department of Gastroenterology, Hepatology, and Nutrition, University of Minnesota, Minneapolis, Minnesota, United States of America, **2** Department of Medicine, Division of Gastroenterology and Hepatology, Bangkok Hospital Pattaya, Chonburi, Thailand, **3** University of Minnesota Medical School, Minneapolis, Minnesota, United States of America, **4** Department of Medicine, University of Minnesota, Minneapolis, Minnesota, United States of America, **5** Gastroenterology Section, Minneapolis VA Health Care System, Minneapolis, Minnesota, United States of America, **6** Clinical and Translational Science Institute, University of Minnesota, Minneapolis, Minnesota, United States of America, **7** Department of Medicine, Division of Pulmonology and Critical Care, University of Minnesota, Minneapolis, Minnesota, United States of America, **8** Department of Pediatrics, Division of Pediatric Gastroenterology, and Hepatology and Nutrition, University of Minnesota, Minneapolis, Minnesota, United States of America, **9** Division of Gastroenterology, Hepatology, and Nutrition, Atrium Health, Wake Forest University, Charlotte, North Carolina, United States of America

‡ These authors are co-first authors on this work.
* nwongjarupong@gmail.com

## Abstract

### Background

Gastrointestinal (GI) complications are the second most common disorders in persons with cystic fibrosis (PwCF). There is limited data on how having a dedicated CF-GI clinic and cystic fibrosis transmembrane conductance regulator (CFTR) modulators may affect rates of GI complications. Our aim was to assess the effect of the CF-GI clinic and CFTR modulators on GI complications with incidence of distal intestinal obstructive syndrome (DIOS).

### Methods

This was a retrospective study of adult PwCF who were seen in a CF-GI clinic from 2000–2023. Comparisons were made between the numbers of admissions and emergency department (ED) visits for DIOS at three years before and after CFTR modulator use and the first CF-GI clinic visit.

**Data availability statement:** All relevant data are within the paper and its Supporting Information files.

**Funding:** This study was financially supported by the National Institutes of Health's National Center for Advancing Translational Sciences in the form of a grant (UM1TR004405) received by TW and AB. This study was financially supported by the Cystic Fibrosis Foundation in the form of a grant (00451A121) received by BM. This study was also financially supported by the Cystic Fibrosis Foundation in the form of a grant (NICE-CF, PROMISE-OB-18, DIGEST 4, and STRONG-CF) received by SS.

**Competing interests:** Sarah Schwarzenberg is a consultant for Uptodate. Baharak Moshiree received Cystic Fibrosis Foundation grant support. Other authors declared no conflict of interest. This does not alter our adherence to PLOS ONE policies on sharing data and materials.

**Abbreviation:** CF, cystic fibrosis; CFTR, cystic fibrosis transmembrane conductance regulator; DIOS, distal intestinal obstruction syndrome; ED, emergency department; GI, gastrointestinal; PwCF, persons with cystic fibrosis.

## Results

Of the 1,076 PwCF identified, 242 were seen in CF-GI clinic. Of this, 126 (52.1%) were female, with a median age of 40 (IQR: 30–47) years. There were 146 (60.3%) with regular use of laxatives. Of the 59 PwCF with CF-GI clinic visits for constipation, hospital admissions decreased in 16, were unchanged in 32, and increased in 11 ($p = 0.402$) while ED visits decreased in 9, remained the same in 40, and increased in 10 ($p = 0.862$). Of the 125 PwCF with CFTR modulator use, DIOS-related hospital admissions decreased in 15 patients, remained unchanged in 89, and increased in 21 ($p = 0.021$) while ED visits were fewer in 8, unchanged in 97, and increased in 20 ($p = 0.587$).

## Conclusion

PwCF had high burden of constipation with a majority of patients regularly using laxatives, and almost half had a history of DIOS. CFTR modulator use and CF-GI clinic were not associated with a decrease of DIOS incidence.

---

## Introduction

Cystic fibrosis (CF) is a multisystem, autosomal recessive genetic disorder caused by mutations or variants in the CF transmembrane conductance regulator (CFTR) gene that encodes an anion channel that controls the movement of chloride, bicarbonate, and water across the epithelium. With the advent of CFTR modulators life expectancy has been increasing, and thus, so have gastrointestinal (GI) manifestations, including gastroesophageal reflux disease, chronic abdominal pain, liver disease, exocrine pancreatic insufficiency, small intestinal bacterial overgrowth, distal intestinal obstruction syndrome (DIOS), and chronic constipation, increasingly being recognized as important complications of CF. Moreover, persons with CF (PwCF) who experience recurrent problems with constipation and DIOS report having a lower quality of life [1].

It is estimated that 10–57% of PwCF report constipation symptoms, and this number may be even as high as 73% [2,3]. The prevalence of DIOS is estimated to be 10–16%, in which most of them had multiple episodes of DIOS in a lifetime [4,5]. However, because of the overlap in clinical presentation of DIOS and constipation this may represent an over- or underestimate of the true prevalence of these conditions. Recent efforts have been made to develop definitive criteria for the diagnosis of these two entities [6]. While DIOS and chronic constipation may present with abdominal pain and bloating, mechanistically, DIOS refers incomplete or complete obstruction within the distal small intestine due to mucus and fecal matter, whereas constipation results from accumulation of stool within the colon [6].

While CFTR modulators are associated with significant improvement in lung function, their effect on GI outcomes is still being studied (NCT04038047). A finding

from a prospective observational study, the PROMISE-GI study, suggested that treatment with the CFTR modulator (elexacaftor/tezacaftor/ivacaftor) led to trivial improvements in GI symptoms (at six months) [7]. However, longer-term follow-up has not been assessed.

In this study, we aimed to evaluate the impact of CFTR modulator therapy and CF-GI clinic referral on the burden of GI complications, specifically DIOS, by analyzing associated health care utilization, including hospital admissions and emergency department (ED) visits.

## Materials and methods

### Study population and data abstraction

We conducted a retrospective review of the medical records of adult PwCF cared for at the University of Minnesota Medical Center in their multidisciplinary CF-GI Clinic. Data were collected for PwCF age 18 or more from January 1 2000 to November 1 2023 who were identified (1) through the ICD-9 and ICD-10 codes for the diagnosis of CF and (2) through the University of Minnesota CF database. The diagnosis of CF was then confirmed by chart review based on the Consensus Guideline from the CF Foundation [8]. Participants were included for analysis if they had been seen or referred to the CF-GI clinic for evaluation of any GI-related symptoms.

Data collected included age, sex, race/ethnicity, CFTR gene mutation, and a history of one of the following: lung transplant, pancreatic insufficiency, and cirrhosis. Information on the history of meconium ileus, DIOS, chronic constipation, gastroparesis, and other gastrointestinal comorbidities were recorded. DIOS was defined based on the ESPGHAN CF Working Group with (1) a short history (days) of abdominal pain or distension or both and; (2) a fecal mass in ileocecum, with or without sign of complete obstruction [9]. For the purpose of our study, we defined constipation as the use of daily laxatives for more than three months, based on prescription data and chart review as using the exact Rome IV criteria was not feasible due to retrospective nature of the study, the long time span of the cohort, and the lack of consistent symptom-based documentation over the years. The use of medications such as CFTR modulators, laxatives, and pharmacologic medications for constipation were also abstracted. A text search for each of the following medications was performed: polyethylene glycol, sennosides, bisacodyl, linaclotide, lubiprostone, plecanatide, and prucalopride. Tenapanor was not included as it was approved by the United States Food and Drug Administration at the same time of the date of last data collection (October 17, 2024). Medication use was confirmed based on clinic notes documenting active use and recorded as prior use or current use. This study was approved by the University of Minnesota Institutional Review Board (STUDY00017426). The Institutional Review Board waived the requirement for informed consent due to minimal risk to the participants. The data was accessed for research propose on November 3 2023. The authors had access to information that could identify individual participants during data collection.

### Statistical analysis

Descriptive data on baseline characteristics of our study cohort were analyzed. Baseline characteristics were reported as mean and standard deviation (SD) or median and interquartile range (IQR) for continuous variables, and percentage for categorical variables. Comparisons were made between numbers of admissions and ED visits during the three years before and after both CFTR modulator initiation and the first CF-GI clinic visit. The number of admissions and ED visits for DIOS were compared between the three years before and after the first CF-GI clinic visit using Wilcoxon signed-rank test. A p-value of less than 0.05 was considered statistically significant, and all reported p-values reflect two-tailed tests. In addition, the utilization of laxatives, secretagogues, and 5-HT4 agonists use were also compared between the two groups. The t-tests were used to compare continuous data, and a Pearson's chi-square or Fischer's exact tests were used to compare the categorical data. All analyses were performed using JMP Pro version 16 software (SAS Institute Inc., Cary, NC).

## Results

### Patient characteristics

Of a total of 1,076 PwCF who were identified as receiving care at our center there were 242 individuals referred to the CF-GI clinic for evaluation of GI symptoms and included in our analysis. Of these, 126/242 (52.1%) PwCF were female, 237/242 (97.9%) were White, the median (IQR) age was 40 (30–47) years, 114/242 (47.1%) PwCF had homozygous delta F508 mutation, 99 (40.9%) PwCF had heterozygous delta F508 mutation, 23/242 (9.5%) PwCF with other varying mutations, and 6/242 (2.5%) had missing information (**Table 1**). Of the 242 PwCF, 159 (65.7%) had or were currently pre-scribed CFTR modulators. And 43/242 (18.2%) PwCF had undergone lung transplant.

Regarding CF-related comorbidities and complications, 216/242 (89.3%) PwCF had pancreatic insufficiency requiring pancreatic enzyme replacement therapy (PERT), 129/242 (53.3%) PwCF had CF-related diabetes, 34/242 (14.1%) PwCF had been diagnosed with gastroparesis by a four-hour standardized gastric emptying scintigraphy, and 16/242 (6.6%) PwCF had been diagnosed with cirrhosis by a hepatologist. Prior history of meconium ileus and DIOS were also analyzed. Fourty-nine of 242 (20.2%) PwCF had a history of meconium ileus as a newborn, of which 31 (63.3%) were treated with surgery. One hundred and sixteen of 242 (47.9%) PwCF had experienced at least one episode of DIOS. The treatment for the first episode of DIOS included medication with polyethylene glycol or N-acetylcysteine (n = 62, 53.4%), gastrografin enema (n = 35, 30.2%), surgery (n = 8, 6.9%), and others (n = 7, 6.0%). The other treatments included self-resolution with hydration (n = 2), nasogastric tube with suction (n = 1), increased PERT (n = 1), and methylnaltrexone (n = 1). One patient developed colon perforation that was treated conservatively (n = 1), and one patient underwent colonoscopy with fecalith removal (n = 1).

### Laxative and prescription medication for constipation in PwCF

Among the 242 PwCF referred to CF-GI clinic, 107 (44.2%) were seen for constipation or recurrent DIOS. With respect to medication use, 146 (60.3%) had documented regularly scheduled laxatives for more than three months. The majority of individuals, 190 (78.5%) used polyethylene glycol as their primary treatment. PwCF who were seen in the CF-GI clinic for constipation were more likely to be prescribed secretagogues or 5-HT4 agonists for management of constipation: 43/107 (40.2%) as compared with 24/135 (17.8%) PwCF not seen in the CF-GI clinic for constipation, $p < 0.001$ (**Table 2**). Among PwCF seen in CF-GI clinic for constipation, linaclotide was the most prescribed medication (n = 27/107, 25.2%). Lubiprostone was the most commonly prescribed medication among the PwCF not seen in CF-GI clinic for constipation (n = 23/135, 17.0%).

### Comparison of numbers of admission and emergency department visits for DIOS before and after CF-GI clinic visit

Of the 242 PwCF seen in the CF-GI clinic, there were 59 PwCF who had adequate data available for the three years prior to and after being seen by a CF-GI in clinic. Thirty-one (53%) were females. There were 36 (61.0%) PwCF with a history of documented DIOS (S1 Data).

To determine if consultation and management by a CF-GI specialist impacted hospital utilization for DIOS, we compared the number of hospital admissions and ED visits for DIOS three years before and after being seen in CF-GI clinic for our cohort of 59 PwCF. We identified 16 PwCF with a decreased number of hospital admissions, 32 PwCF with an unchanged number of admissions, and 11 PwCF with an increased number of admissions (**Fig 1**). With respect to the number of ED visits, we identified nine PwCF with a decreased number of ED visits, 40 with an unchanged number, and 10 patients with an increased number of ED visits (**Fig 2**). The change in the number of DIOS admissions in the hospital versus the ED were not statistically different between the three years before and three years after a CF-GI clinic visit ($p = 0.402$, and 0.862, respectively).

**Table 1. Patient characteristics of individuals referred to or evaluated by a cystic fibrosis-gastrointestinal clinic.**

| Characteristics | N=242 (%) |
|---|---|
| Sex, female % | 126 (52.1%) |
| Age, year (median (interquartile range)) | 40 (30-47) |
| Race | |
| White | 237 (97.9%) |
| Black | 2 (0.8%) |
| Asian/Pacific Islander | 2 (0.8%) |
| Missing | 1 (0.4%) |
| Ethnicity | |
| Non-Hispanic, % | 150 (62.0%) |
| Hispanic | 1 (0.4%) |
| Missing, % | 91 (37.6%) |
| CFTR gene mutation | |
| F508 deletion homozygous | 114 (47.1%) |
| F508 deletion heterozygous | 99 (40.9%) |
| Other | 23 (9.5%) |
| No information | 6 (2.5%) |
| CFTR modulator use | 159 (65.7%) |
| Lung transplant | 43 (18.2%) |
| Exocrine pancreatic insufficiency | 216 (89.3%) |
| Cystic fibrosis-related diabetes | 129 (53.3%) |
| Gastroparesis | 34 (14.1%) |
| Cirrhosis, seen by liver clinic | 16 (6.6%) |
| History of meconium ileus | 49 (20.2%) |
| Surgery for meconium ileus | 31 (12.8%) |
| DIOS history | 116 (47.9%) |
| Treatment for first DIOS presentation | |
| Medication/polyethylene glycol or N-acethylcysteine | 62 (53.4%) |
| Gastrografin enema | 35 (30.2%) |
| Surgery | 8 (6.9%) |
| Others | 7 (6.0%) |
| Missing | 4 (3.4%) |
| Any history of surgery for DIOS | 1 (4.1%) |
| Any history of surgery for small bowel obstruction | 8 (3.3%) |
| History of bowel resection, not for DIOS/constipation/ meconium ileus | 28 (11.6%) |

Abbreviation: CFTR cystic fibrosis transmembrane conductance regulator; DIOS distal intestinal obstruction syndrome

Of the 59 PwCF in the CF-GI clinic cohort, there were 29 PwCF who started CFTR modulator during the six-year period of study; of which 10 were prior to the CF-GI clinic first visit, and 19 were after the CF-GI clinic first visit. There were five PwCF who underwent lung transplant, of which four underwent lung transplant prior to the CF-GI clinic first visit, and one underwent lung transplant after the CF-GI clinic first visit.

**Table 2. Comparison of medication utilization in persons with cystic fibrosis seen and not seen by a cystic fibrosis-gastrointestinal clinic for constipation.**

| | Total cohort n = 242 | Seen in CF-GI clinic for constipation n = 107 | Seen in CF-GI for symptoms other than constipation n = 135 |
|---|---|---|---|
| Regular laxative use > 3 months | 146 (60.3%) | 86 (80.4%) | 60 (44.4%) |
| Prescription medications, any | 67 (27.7%) | 43 (40.2%) | 24 (17.8%) |
| One medication | 49 (20.2%) | 27 (25.2%) | 22 (16.3%) |
| Two medications | 16 (6.6%) | 14 (13.1%) | 2 (1.5%) |
| Three medications | 2 (0.8%) | 2 (1.9%) | 0 |
| **Secretagogues** | | | |
| Linaclotide | 29 (12.0%) | 27 (25.2%) | 2 (1.5%) |
| Lubiprostone | 47 (19.4%) | 24 (22.4%) | 23 (17.0%) |
| Plecanatide | 4 (1.7%) | 3 (2.8%) | 1 (0.7%) |
| **5 HT4 agonist** | | | |
| Prucalopride | 7 (2.9%) | 7 (6.5%) | 0 (0%) |
| **Over the counter laxatives** | | | |
| Lactulose | 32 (29.3%) | 20 (18.7%) | 12 (8.9%) |
| Polyethylene glycol | 190 (78.5%) | 95 (88.8%) | 95 (70.4%) |
| Sennosides | 103 (42.6%) | 53 (49.5%) | 50 (37.0%) |
| Bisacodyl | 39 (16.1%) | 23 (21.4%) | 16 (11.8%) |

Abbreviation: CF-GI cystic fibrosis-gastrointestinal

### Comparison of numbers of admission and ED visits for DIOS before and after CFTR modulator initiation

Of the 242 PwCF in our cohort, there were 159 (65.7%) PwCF who used CFTR modulators. The CFTR modulators included elexacaftor/tezacaftor/ivacaftor (n = 71), lumacaftor/ivacaftor (n = 60), tezacaftor/ivacaftor (n = 19), and ivacaftor (n = 9). Of these, 125 (51.7%) PwCF who had adequate follow-up of three years before and after CFTR modulator initiation with the baseline characteristics was provided in S2 Data. Sixty-four (51.2%) were female. There were 61 (48.8%) PwCF with any history of documented DIOS in their lifetime.

Among individuals taking CFTR modulators, the number of hospital admissions for DIOS three years before and after were collected, and 15 PwCF had decreased number of admissions; in 89 PwCF, the number of admissions was unchanged, and 21 PwCF had increased number of admissions (Fig 3). For the number of ED visits for DIOS three years before and after the CFTR modulator initiation, there were eight PwCF with decreased number of ED visits, 97 with unchanged; and 20 patients with increased number of ED visits, respectively (Fig 4). There was an increased number of admissions for DIOS after CFTR modulator initiation ($p = 0.021$), but no statistical difference in the number of ED visits ($p = 0.587$). The changes in number of admissions and ED visits by types of CFTR modulators was provided in S3 Data.

Of the 125 PwCF in the CFTR modulator cohort, there were 38 PwCF who had the first CF-GI clinic during the six-year period of study of which 19 were before the CFTR modulator initiation, and 19 were after CFRT modulator initiation. There were three PwCF who underwent lung transplant during the six-year period of study of which all underwent transplant after CFTR modulator initiation.

## Discussion

GI manifestations of CF, including constipation and DIOS, represent a significant source of morbidity and negatively impact quality of life among PwCF. The results of our study help characterize the prevalence, treatment patterns in

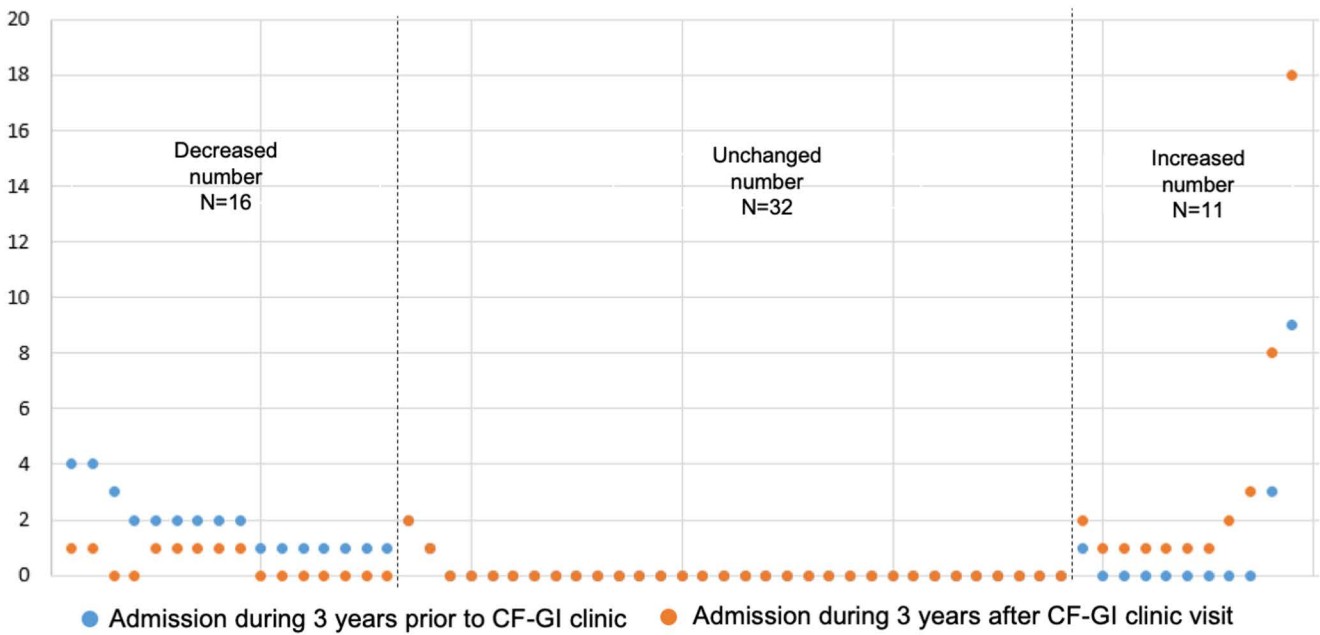

**Fig 1. Dot plot of number of admissions for distal intestinal obstruction syndrome (DIOS) three years before and three years after being seen in cystic fibrosis-gastrointestinal (CF-GI) clinic (n = 59).**

multidisciplinary CF clinics, and hospital utilization associated with these two common GI complications associated with CF.

Our results show that the most common agent used to treat constipation in PwCF is polyethylene glycol, likely due to its safety profile, ease of dosing, long-term data especially in children, where until recently many of the prescription medications were not FDA-approved. Additionally, polyethylene glycol can be titrated, or adjusted by the patient depending on their symptoms. In the GALAXY study, a recent large multi-center prospective study based on a validated questionnaire, polyethylene glycol was also the most prescribed medication in adults and pediatric patients [1]. In our study, lubiprostone was the most common pharmacologic medication used to treat constipation, primarily prescribed by the patient's CF pulmonologist. However, linaclotide was the prescribed medication of choice for constipation when PwCF were referred to specialized CF-GI clinic.

The primary mechanism of action of linaclotide and part of the mechanism of action of lubiprostone are through activation of the CFTR. A prior *in-vitro* study of the intestinal epithelium of PwCF with homozygous F508 deletion suggested that the effect of lubiprostone was compromised in PwCF given the decreased to non-function of CFTR chloride channel [10]. However, a study of F508del mutant mice showed that both linaclotide and lubiprostone increased intestinal transit time and intestinal fluid through inhibiting sodium/hydrogen 3 exchanger [11,12]. This sodium/hydrogen 3 exchanger is the main target for tenapanor, which is potentially an alternative mechanism of action for both lubiprostone and linaclotide in PwCF [11,12]. Our study did not have any patients on tenapanor as it was approved by the United States Food and Drug Administration at the same time of the date of last data collection. In the setting of concomitant use with CFTR modulator, lubiprostone was shown to have synergistic effect with CFTR modulators to increase chloride secretion in the nasal epithelium, but data in GI tract remain limited [13]. Studies evaluating the clinical effectiveness of intestinal secretagogues in PwCF (especially in individuals using CFTR modulators) are desperately needed. There was only one pilot clinical study of lubiprostone in PwCF that showed improved overall symptoms of constipation based on Patient Assessment of

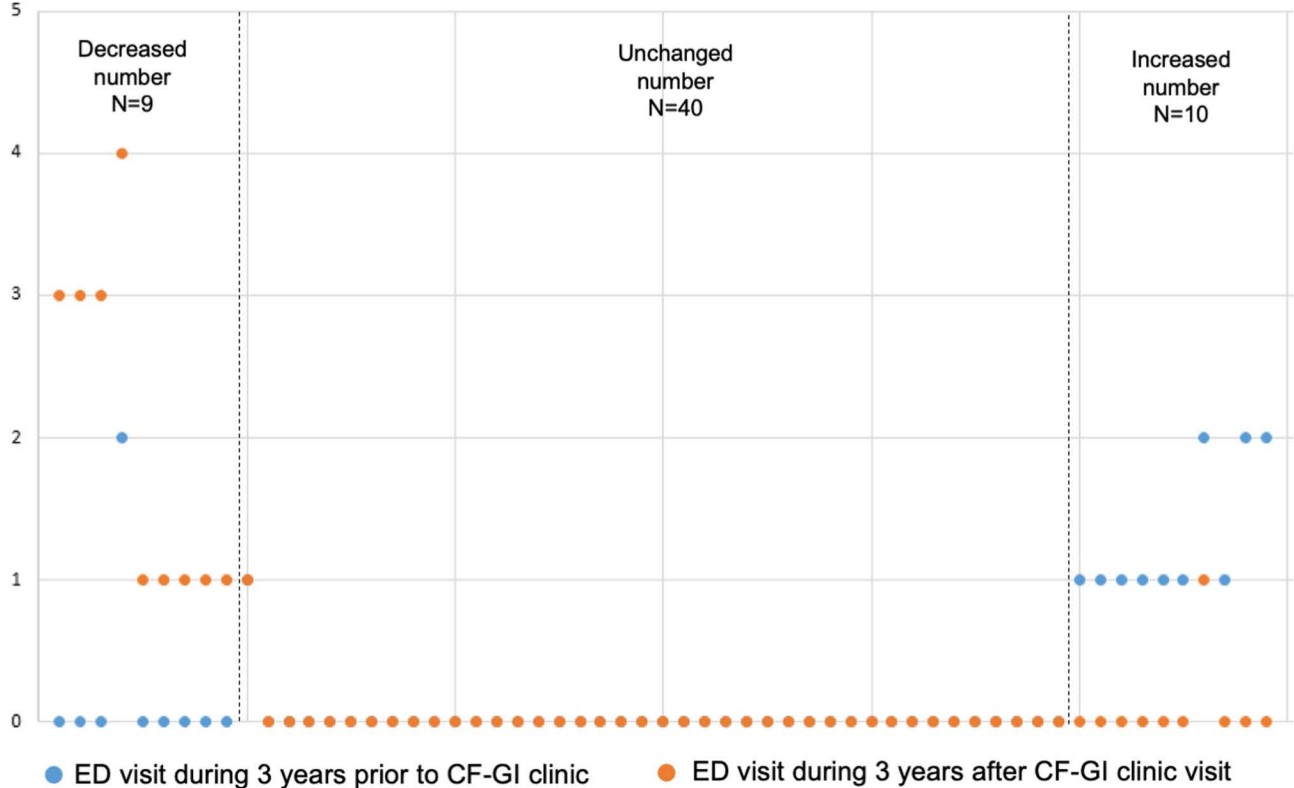

**Fig 2. Dot plot of number of emergency department (ED) visits for distal intestinal obstruction syndrome three years before and three years after being seen by cystic fibrosis-gastrointestinal clinic (n = 59).**

Constipation-Symptoms score, but not in the number of spontaneous bowel movements or with the Bristol Stool Score [14]. Based on a Cochrane review, the only available randomized cross-over study on DIOS prevention is with using cisapride in only 17 participants, and this study did not show any benefit from the medication [15,16].

Despite collaboration with a specialized CF-GI clinic, outcomes of PwCF three years before and after seeing a CF gastroenterologist, there was no decrease in the number of hospital admissions for DIOS or the number of ED admissions. One possibility for this finding is that a three year follow up period is too short to demonstrate improvement as it could take time to find a stable medication regimen to treat patients who have constipation or DIOS. It is also possible that other factors, such as lung transplant affected the GI complications. In a prior study of PwCF who underwent transplant, 20% developed DIOS that occurred immediately during post-operative period [17]. Also, the subset of individuals referred to the CF-GI clinic represent a difficult-to-treat population with more severe disease. Although we observed no significant change in health care utilization after CF-GI referral, we acknowledge the possibility that referral may have slowed progression or mitigated further worsening of symptoms. In the absence of a comparable non-referred control group, we were unable to assess this counterfactual. Our findings also align with prior systematic reviews suggesting that currently available therapies for DIOS and constipation offer limited benefit, and our data may reflect the lack of highly effective treatment options for these complications despite specialized care [15].

In a subset of PwCF in this study, there was an increase in DIOS-related admissions and ED visits after the initiation of CFTR modulator (over three years). This is consistent with prior studies that show that the use of CFTR modulator did not

Number of admissions for DIOS

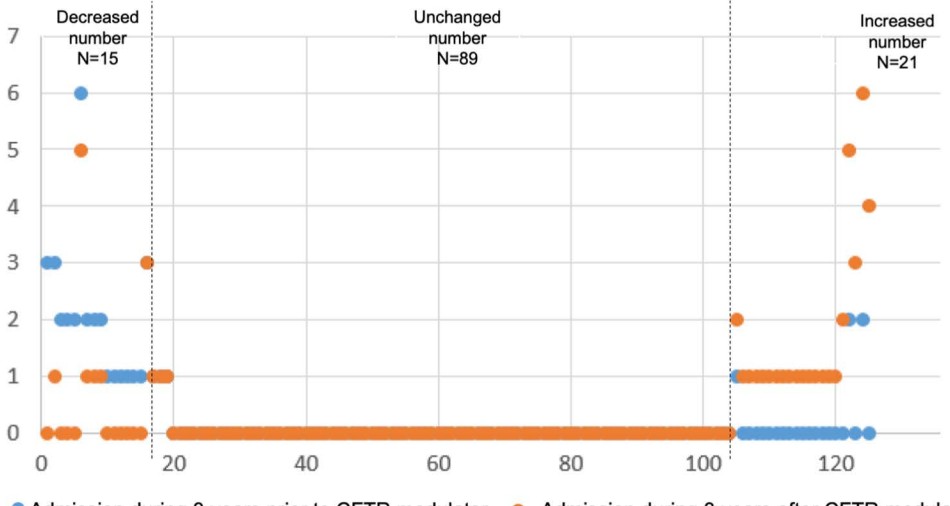

**Fig 3. Dot plot of number of admissions for distal intestinal obstruction syndrome (DIOS) three years before and three years after cystic fibrosis transmembrane conductance regulator (CFTR) modulator initiation (n = 125).**

## Number of ED visits for DIOS

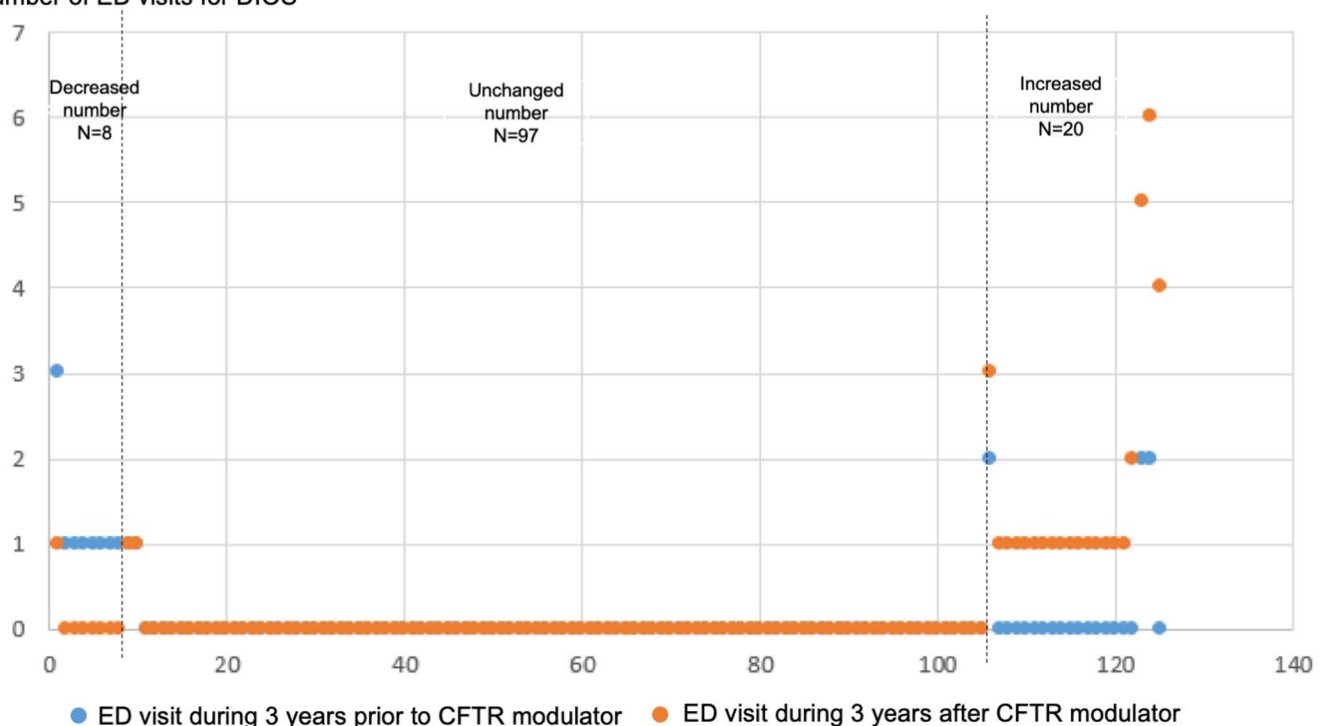

**Fig 4. Dot plot of number of emergency department (ED) visits for distal intestinal obstruction syndrome (DIOS) three years before and three years after cystic fibrosis transmembrane conductance regulator (CFTR) modulator initiation (n = 125).**

impact patient-reported GI symptom scores [1,7]. The median time from the CFTR modulator to the best pulmonary function test takes up to 100 days [18]. Currently, there are no data on how long it would take for the CFTR modulator to take effect on the GI tract, and it could take a longer duration than the three-year follow-up time in our study. There is also a plausible mechanism that the initiation of CFTR modulator could cause hydration of the viscous mucous and subsequently detach the fecal matter from the luminal wall and cause DIOS [19].

In our study, DIOS contributed to admissions and ED visits for more than a quarter of the PwCF in the cohort. Only one prior study had quantified health care utilization among PwCF. In this prior study of 345 PwCF, the primary factors associated with health care utilization were pancreatic insufficiency, being positive for *Pseudomonas aeruginosa*, and CF-related diabetes [20]. DIOS was not identified as a factor to be significantly related to high cost of health care utilization, but only 8.7% of individuals had a history of DIOS [20]. Furthermore, this prior study predated the use of CFTR modulators (conducted from 2009 to 2017), and the majority of health care costs were related to pulmonary complications. Of note, the prevalence of DIOS in our study is much higher at 47.9% suggesting a more significant burden of GI-related diseases.

There were several limitations to our study. First, this was a retrospective cohort study with limited information on patient reported outcomes. The outcomes used in this study were the number of admissions and ED visits that might not have accounted for PwCF with milder symptoms of DIOS and constipation. Secondly, detailed information on access to pharmacological therapies, adherence, and use of prescribed therapies was not available. Additionally, our analysis lacked a comparison group of PwCF not referred to a CF-GI clinic, which limits our ability to determine whether outcomes would have been worse without specialist intervention. Lastly, the findings pre and post CFTR use with respect to DIOS and constipation risk may solely be due to lack of longer-term follow-up.

## Conclusion

Constipation and DIOS were prevalent in PwCF. The most commonly used laxative was polyethylene glycol. The most commonly prescribed medication for constipation was lubiprostone, and when PwCF were referred to CF-GI clinic, the most prescribed medication was linaclotide. Referral to a CF-GI specialist was not associated with a decrease in DIOS-associated hospital admissions and ED visits as would be expected but this may be due to insurance restrictions and prior authorization requirements limiting prescription medication coverage. Additionally, perhaps a longitudinal study for a longer duration of observation is necessary to find decreased hospitalization rates and ED visits in these patients. Additionally, CFTR modulator use over 3 years was also not associated with decrease in DIOS-related hospital admissions and ED visits. GI complications in PwCF require longer term studies so we can further characterize and reduce health care utilization and costs and improve outcomes and quality of life in PwCF.

## Supporting information

**S1 Data. Characteristics of patients seen in cystic fibrosis-gastrointestinal clinic for comparison of three years before and after cystic fibrosis-gastrointestinal clinic.**
(DOCX)

**S2 Data. Characteristics of patients seen in cystic fibrosis-gastrointestinal clinic for comparison of pre- and post-cystic fibrosis transmembrane conductance regulator (CFTR) modulator initiation.**
(DOCX)

**S3 Data. Changes in number of admissions and ED visits before and after CFTR modulator initiation by types of CFTR modulators.**
(DOCX)

**S4 Data. Study data set.**
(XLSX)

## Author contributions

**Conceptualization:** Nicha Wongjarupong, Malique F Delbrune, Jameel Alp, Daphne M Moutsoglou, Talia Wiggen, Ashley Benner, Joanne L Billings, Sarah J Schwarzenberg, Baharak Moshiree, Shahnaz Sultan.

**Data curation:** Nicha Wongjarupong, Malique F Delbrune, Jameel Alp, Talia Wiggen, Ashley Benner, Shahnaz Sultan.

**Formal analysis:** Nicha Wongjarupong.

**Investigation:** Nicha Wongjarupong, Malique F Delbrune, Shahnaz Sultan.

**Methodology:** Nicha Wongjarupong, Malique F Delbrune.

**Resources:** Daphne M Moutsoglou.

**Software:** Talia Wiggen, Ashley Benner.

**Supervision:** Daphne M Moutsoglou, Joanne L Billings, Jordan M Dunitz, Sarah J Schwarzenberg, Baharak Moshiree, Shahnaz Sultan.

**Visualization:** Daphne M Moutsoglou, Joanne L Billings, Jordan M Dunitz, Baharak Moshiree, Shahnaz Sultan.

**Writing – original draft:** Nicha Wongjarupong, Malique F Delbrune, Jameel Alp, Daphne M Moutsoglou.

**Writing – review & editing:** Nicha Wongjarupong, Malique F Delbrune, Jameel Alp, Daphne M Moutsoglou, Talia Wiggen, Ashley Benner, Joanne L Billings, Jordan M Dunitz, Sarah J Schwarzenberg, Baharak Moshiree, Shahnaz Sultan.

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
