## [Decision Letter · Decision Letter 0]

Dear Dr. Wongjarupong,

Thank you for submitting your manuscript to PLOS ONE. After careful consideration, we feel that it has merit but does not fully meet PLOS ONE’s publication criteria as it currently stands. Therefore, we invite you to submit a revised version of the manuscript that addresses the points raised during the review process.

**ACADEMIC EDITOR comments:**

In addition to the 2 reviewer comments below, please address these 2 minor issues:in the results section page 8, it is written: "PwCF who were seen in the CF-GI clinic for constipation were more likely to be prescribed secretagogues or 5-HT4 agonists for management of constipation: 43/107 (40.2%) as compared with 24/135 (17.8%) PwCF not seen in the CF-GI clinic, p <0.001 [Table 2]" - i think this should be 'not seen in the CF-GI clinic FOR CONSTIPATION"in results, page 10, it's written: " Of these, **t1`25 (51.7%)** PwCF who had adequate follow-up of three years before and after CFTR modulator initiation with the baseline characteristics provided in Supplementary data 2." - I believe this should be "125 (51.7%).

We look forward to receiving your revised manuscript.

Kind regards,

Alyssa Kriegermeier

Academic Editor

PLOS ONE

Journal Requirements:

Sarah Schwarzenberg is a consultant for Uptodate.

Baharak Moshiree received Cystic Fibrosis Foundation grant support; Bausch Pharma grant support; and Salix grant support.

Other authors declared no conflict of interest.

4. Your abstract cannot contain citations. Please only include citations in the body text of the manuscript, and ensure that they remain in ascending numerical order on first mention.

5. Please remove all personal information, ensure that the data shared are in accordance with participant consent, and re-upload a fully anonymized data set.

Reviewers' comments:

Reviewer's Responses to Questions

**Comments to the Author**

1. Is the manuscript technically sound, and do the data support the conclusions?

Reviewer #1: Partly

Reviewer #2: Yes

2. Has the statistical analysis been performed appropriately and rigorously?

Reviewer #1: Yes

Reviewer #2: Yes

3. Have the authors made all data underlying the findings in their manuscript fully available?

Reviewer #1: No

Reviewer #2: Yes

4. Is the manuscript presented in an intelligible fashion and written in standard English?

Reviewer #1: Yes

Reviewer #2: Yes

Reviewer #1: Well prepared manuscript

This is an understudied topic area and the reviewers provide helpful, relevant and new data which somewhat surprised this reviewer who had assumed that the use of CFTR modulator therapy would have some impact upon DIOS and perhaps constipation

The topic area is difficult to study as DIOS increases with age, and therefore there is a possibility that some modest benefit is being derived but the findings are, in my view, an important addition to the literature and will be helpful in discussions with patients

The authors acknowledge the most important limitation of the study; namely that ‘the subset of individuals referred to the CF-GI clinic represent a difficult-to-treat population with more severe disease’

In the abstract the provision of nonsignificant p values greater than 0.05 is not required or, in the opinion of this reviewer helpful: they are simply nonsignificant. It would be more helpful to see the number of episodes reported (median).

Whilst they conclude that referral to a CF-GI clinic does not result in better outcomes for patients there are two important considerations:

1. Would the natural history be worse for those never referred? i.e. is there an inexorable increase in gut symptoms that is slowed or halted with optimal treatment.

2. Does this simply confirm the results of the systematic reviews showing that no treatments that are currently available for use are effective.

My biggest grumble about the proposed paper is that I find the presentation of the results very difficult to follow. Surely the number of ED visits for DIOS over 3 years before and 3 years after initiation of CFTR modulator therapy is a number, which could be summarized as a median and IQR.

There are other concerns about the figures. For instance: what does ‘similar number’ mean. Does it mean ‘same number’?

They are of low quality

They have grammatical errors i.e. Number of admission for DIOS rather than Number of admissions for DIOS

Some revision of these and careful thought about how to present the data is required.

Minor typographical error on line 183 with no space between the words identified and 16

Minor typo line 203, of these t1`25 should be 125 (51.7%)

Minor typo line 280: ‘aeruginosa’ needs to be in italics

Minor typo line 282: no space between DIOS and (20)

Reviewer #2: This retrospective study by Wongjarupong N, et al. looking at the impact of CF-GI clinics and HEMT on the incidence of DIOS and constipation in PwCF adds significant insight to our current understanding of this issue. This study raises further questions regarding proper identification of DIOS and constipation patients, long-term effects of HEMT on GI symptoms, proper treatment of constipation, and utility of specialized CF-GI clinics. These questions build a framework for future studies on the subject.

Comments:

1. The aim stated in the introduction (characterizing DIOS/constipation burden, healthcare utilization) seems to vary from that which is described in the title and conclusion (impact of CF-GI clinics and HEMT on DIOS/constipation).

2. I think the 3-year timepoint before and after intervention is very reasonable, and should give enough time to see differences in the stated outcomes.

3. While prior lung transplant may be important for DIOS, particularly immediately post-transplant, there are no data in this study regarding timing of the prior transplant. It might be better to note how many of those patients had lung transplant within the 6-year timeframe of the study.

4. It sounds as if only patients who were referred to or seen by CF-GI were included in the study. Do you have data regarding incidence of DIOS before and after HEMT in patients who may not have been referred to GI?

5. Timing of initiation of HEMT and first visit to CF-GI could have confounded the results. It may be worth including data regarding initiation of HEMT in relation to CF-GI visit. With the current data, it is difficult to tease out the impact of HEMT vs. CF-GI on your reported outcomes.

6. Your study defined constipation based only on daily laxative use for > 3 years. The standard definition of constipation is typically symptom-based. Your stringent definition may not be the best to look at differences in constipation outcomes with your interventions. While constipation symptoms may have been significantly better with CF-GI intervention (i.e. being started on effective medications), the patient would be defined as still having constipation if they continued their constipation meds even though their symptoms could have resolved.

7. Do you have enough numbers of patients to break improvement with intervention down by type of HEMT? The more effective modulators may be more of less beneficial for GI symptoms.

**Do you want your identity to be public for this peer review?** For information about this choice, including consent withdrawal, please see our Privacy Policy

Reviewer #1: **Yes: ** Professor Will Carroll

Reviewer #2: No

---

## [Author Response · Author response to Decision Letter 1]

31 May 2025

We would like to thank the reviewers and the editorial team for their thoughtful, constructive, and insightful feedback on our manuscript. Below, we have provided detailed responses to each point raised and indicated the changes made in the revised version.

Academic editor

1. In the results section page 8, it is written: ‘PwCF who were seen in the CF-GI clinic for constipation were more likely to be prescribed secretagogues or 5-HT4 agonists for management of constipation: 43/107 (40.2%) as compared with 24/135 (17.8%) PwCF not seen in the CF-GI clinic, p <0.001 [Table 2]’ - i think this should be 'not seen in the CF-GI clinic FOR CONSTIPATION’. In results, page 10, it's written: ‘Of these, t1`25 (51.7%) PwCF who had adequate follow-up of three years before and after CFTR modulator initiation with the baseline characteristics provided in Supplementary data 2.’ - I believe this should be ‘125 (51.7%).

Response: Thank you for pointing out these issues, and we apologize for the oversight. We have made the corrections in the manuscript: The phrase "not seen in the CF-GI clinic" has been updated to "not seen in the CF-GI clinic for constipation" on page 8 as suggested. The typo on page 10 has been corrected to read "125 (51.7%)" instead of "t1`25."

Reviewer 1

1. “In the abstract the provision of nonsignificant p values greater than 0.05 is not required or, in the opinion of this reviewer helpful: they are simply nonsignificant. It would be more helpful to see the number of episodes reported (median).”

Response: Thank you for your helpful comment regarding the reporting of nonsignificant p-values in the abstract. We have revised the Results section to emphasize directional changes in hospital admissions and ED visits by providing categorical summaries (i.e., decreased/unchanged/ increased), rather than highlighting nonsignificant p-values. We hope this approach more effectively conveys the clinical patterns observed in our cohort, in line with your suggestion. We also considered the inclusion of median values; however, as the number of DIOS-related episodes was zero for many patients, we thought median values may offer limited interpretability, and therefore, we opted for categorical reporting to better reflect variability across the cohort. We appreciate your insightful feedback and hope these revisions enhance the clarity and impact of the abstract.

2. Whilst they conclude that referral to a CF-GI clinic does not result in better outcomes for patients there are two important considerations:

1. Would the natural history be worse for those never referred? i.e. is there an inexorable increase in gut symptoms that is slowed or halted with optimal treatment.

2. Does this simply confirm the results of the systematic reviews showing that no treatments that are currently available for use are effective.

Response: We appreciate this insightful comment.

1. As the reviewer notes, it is possible that referral to a CF-GI clinic may have slowed progression or mitigated worsening of gastrointestinal symptoms, but we were unfortunately unable to capture due to the retrospective nature of our study and lack of a comparable non-referred control group. We now acknowledge this in the Discussion section as a limitation and include the possibility that outcomes may have been worse without referral.

2. We also appreciate the reviewer’s second point regarding the broader treatment landscape. Our findings may indeed reflect the limited efficacy of currently available therapies for DIOS and constipation in cystic fibrosis, as suggested in systematic reviews. We have added language in the Discussion to place our findings within this context.

3. My biggest grumble about the proposed paper is that I find the presentation of the results very difficult to follow. Surely the number of ED visits for DIOS over 3 years before and 3 years after initiation of CFTR modulator therapy is a number, which could be summarized as a median and IQR. There are other concerns about the figures. For instance: what does ‘similar number’ mean. Does it mean ‘same number’? They have grammatical errors i.e. Number of admission for DIOS rather than Number of admissions for DIOS. Some revision of these and careful thought about how to present the data is required.

Response: Thank you for your thoughtful and constructive feedback regarding the presentation of our results. We appreciate your concern about the clarity and readability of the data, and we have carefully revised the figures in response.

To improve clarity, we have replaced ambiguous terms such as “similar number” with more precise language (e.g., “unchanged”) and corrected grammatical errors, including “number of admission” to “number of admissions.” Regarding your suggestion to report the number of DIOS-related ED visits using medians and interquartile ranges, we gave this careful consideration. However, due to the high proportion of patients with zero events both before and after CFTR modulator initiation, the median value was often zero, which we felt limited its ability to meaningfully capture variability in the data. To better reflect clinically relevant patterns, we chose to present categorical summaries (i.e., decreased/unchanged/increased), which we believe more effectively communicate the observed trends across the cohort. We are grateful for your insights, which helped strengthen the clarity and rigor of the manuscript, and we hope the revised version better reflects these improvements.

4. Minor typographical error on line 183 with no space between the words identified and 16.

Minor typo line 203, of these t1`25 should be 125 (51.7%)

Minor typo line 280: ‘aeruginosa’ needs to be in italics

Minor typo line 282: no space between DIOS and (20)”

Response: Thank you for noting these errors, and we apologize for the oversight. We have corrected all identified issues as follows: added a space between “identified” and the citation, corrected “t1`25” to “125 (51.7%)”, italicized aeruginosa, and inserted a space between “DIOS” and the citation.

Reviewer 2

1. This retrospective study by Wongjarupong N, et al. looking at the impact of CF-GI clinics and HEMT on the incidence of DIOS and constipation in PwCF adds significant insight to our current understanding of this issue. This study raises further questions regarding proper identification of DIOS and constipation patients, long-term effects of HEMT on GI symptoms, proper treatment of constipation, and utility of specialized CF-GI clinics. These questions build a framework for future studies on the subject.

Response: Thank you very much for your thoughtful and encouraging feedback! We greatly appreciate your recognition of the significance of our study in contributing to the care of individuals with cystic fibrosis. We also appreciate the valuable ideas you raised, and we hope to build on these ideas in future work and continue exploring these important questions.

2. The aim stated in the introduction (characterizing DIOS/constipation burden, healthcare utilization) seems to vary from that which is described in the title and conclusion (impact of CF-GI clinics and HEMT on DIOS/constipation).

Response: Thank you for this helpful observation. We agree that consistency in framing the study’s aim is important. To address this, we revised the Introduction to better reflect the primary objectives of the study as stated in the title and conclusion. Namely, “In this study, we aimed to evaluate the impact of CFTR modulator therapy and CF-GI clinic referral on the burden of GI complications, specifically DIOS and constipation, by analyzing associated health care utilization, including hospital admissions and emergency department (ED) visits, as well as reviewing medical charts in PwCF.”

3. I think the 3-year timepoint before and after intervention is very reasonable, and should give enough time to see differences in the stated outcomes.

Response: Thank you for your supportive comment regarding the 3-year time frame, and we appreciate your validation of this approach.

4. While prior lung transplant may be important for DIOS, particularly immediately post-transplant, there are no data in this study regarding timing of the prior transplant. It might be better to note how many of those patients had lung transplant within the 6-year timeframe of the study.

Response: Thank you for the insightful comment. We have looked into our data and added the information on the number of patients who had lung transplant within the 6-year period of study for both CFTR modulator and CF-GI clinic analyses. Of the 59 PwCF in the CF-GI clinic cohort, there were 5 PwCF who underwent lung transplant of which 4 underwent lung transplant prior to the CF-GI clinic first visit and 1 underwent lung transplant after the CF-GI clinic first visit. Of the 125 PwCF in the CFTR modulator cohort, there were 3 PwCF who underwent lung transplant during the 6-year period of study of which all underwent transplant after CFTR modulator initiation. This information was added to the Result section.

5. It sounds as if only patients who were referred to or seen by CF-GI were included in the study. Do you have data regarding incidence of DIOS before and after HEMT in patients who may not have been referred to GI?

Response: Thank you for this important question. While our study focused on the cohort of 242 PwCF referred to the CF-GI clinic, we do have a larger group of 834 PwCF who were not seen in the CF-GI clinic during the study period. However, detailed chart review for DIOS episodes in that broader population was beyond the scope of this project, which required several years of manual data collection even within the referred cohort. We agree that examining DIOS incidence in the non-referred population would be a valuable direction for future research and could provide complementary insights into broader trends following HEMT initiation.

6. Timing of initiation of HEMT and first visit to CF-GI could have confounded the results. It may be worth including data regarding initiation of HEMT in relation to CF-GI visit. With the current data, it is difficult to tease out the impact of HEMT vs. CF-GI on your reported outcomes.

Response: Thank you for reviewer’s thoughtful comment. We have explored our data for this specific question. Of the 59 PwCF in the CF-GI clinic cohort, there were 29 PwCF who started CFTR modulator during the 6-year period of study; of which 10 were prior to the CF-GI clinic first visit and 19 were after the CF-GI clinic first visit. Of the 125 PwCF in the CFTR modulator cohort, there were 38 PwCF who had the first CF-GI clinic during the 6-year period of study of which 19 were before the CFTR modulator initiation, and 19 were after CFRT modulator initiation. This information was added to the Result section.

7. Your study defined constipation based only on daily laxative use for > 3 years. The standard definition of constipation is typically symptom-based. Your stringent definition may not be the best to look at differences in constipation outcomes with your interventions. While constipation symptoms may have been significantly better with CF-GI intervention (i.e. being started on effective medications), the patient would be defined as still having constipation if they continued their constipation meds even though their symptoms could have resolved.

Response: Thank you very much for this insightful comment. We recognize that this may have been unclear in the original text, and we have revised the wording to clarify in the Methods section. We agree that a symptom-based definition of constipation, such as the Rome IV criteria, is the standard. However, due to the retrospective nature of our study, the long span of the cohort, and limited symptom documentation, applying the exact Rome IV criteria was not feasible. As such, we selected a threshold of daily laxative use for more than three months, guided by the chronicity criteria in Rome IV (which require symptoms to be present for at least 3–6 months). We felt this approach would allow for the most consistent identification of individuals with clinically significant constipation while acknowledging its limitations. We have clarified this definition.

8. Do you have enough numbers of patients to break improvement with intervention down by type of HEMT? The more effective modulators may be more of less beneficial for GI symptoms.

Response: Of the 125 patients in the CFTR modulator cohort, there were 49 lumacaftor/ivacaftor, 47 elexacaftor/tezacaftor/ivacaftor, 18 tezacaftor/ivacaftor, 8civacaftor, and 3 with missing information. The table was added as Supplementary data 3.

We look forward to an expeditious and hopefully positive review of the manuscript. We thank the editors and reviewers in advance for their time and effort in procession and reviewing our work.

Sincerely,

Nicha Wongjarupong, MD

---

## [Decision Letter · Decision Letter 1]

Effect of Cystic Fibrosis Transmembrane Conductance Regulator Modulator and Dedicated Cystic Fibrosis Gastrointestinal Clinic Visits on the Incidence of Distal Intestinal Obstructive Syndrome in Persons with Cystic Fibrosis

PONE-D-25-14674R1

Dear Dr. Wongjarupong,

We’re pleased to inform you that your manuscript has been judged scientifically suitable for publication and will be formally accepted for publication once it meets all outstanding technical requirements.

Kind regards,

Alyssa Kriegermeier

Academic Editor

PLOS ONE

Additional Editor Comments (optional):

Reviewers' comments:

Reviewer's Responses to Questions

**Comments to the Author**

Reviewer #1: All comments have been addressed

Reviewer #2: All comments have been addressed

2. Is the manuscript technically sound, and do the data support the conclusions?

Reviewer #1: Yes

Reviewer #2: Yes

3. Has the statistical analysis been performed appropriately and rigorously?

Reviewer #1: I Don't Know

Reviewer #2: Yes

4. Have the authors made all data underlying the findings in their manuscript fully available?

Reviewer #1: No

Reviewer #2: Yes

5. Is the manuscript presented in an intelligible fashion and written in standard English?

Reviewer #1: Yes

Reviewer #2: Yes

Reviewer #1: Thank you for carefully addressing the comments. The paper addresses an important clinical issue and the results are somewhat surprising. They should act as a stimulus for other centres to carefully address whether CFTR modulators have an impact upon GI symptoms. And if so, to what extent.

Reviewer #2: From my standpoint, all reviewer critiques have been adequately addressed. I am comfortable with this manuscript being accepted for publication.

**Do you want your identity to be public for this peer review?** For information about this choice, including consent withdrawal, please see our Privacy Policy

Reviewer #1: **Yes: ** W D CARROLL

Reviewer #2: No

---

## [Editor Report · Acceptance letter]

PONE-D-25-14674R1

PLOS ONE

Dear Dr. Wongjarupong,

I'm pleased to inform you that your manuscript has been deemed suitable for publication in PLOS ONE. Congratulations! Your manuscript is now being handed over to our production team.

Kind regards,

on behalf of

Dr. Alyssa Kriegermeier

Academic Editor

PLOS ONE